# Phytochemical Analysis and Biological Activities of Ripe Fruits of Mistletoe (*Psittacanthus calyculatus*)

**DOI:** 10.3390/plants12122292

**Published:** 2023-06-12

**Authors:** Zaida Ochoa-Cruz, Jorge Molina-Torres, María V. Angoa-Pérez, Jeanette G. Cárdenas-Valdovinos, Ignacio García-Ruiz, José A. Ceja-Díaz, José O. Bernal-Gallardo, Hortencia G. Mena-Violante

**Affiliations:** 1Instituto Politécnico Nacional, Departament of Reserch, CIIDIR IPN Unidad Michoacán, Jiquilpan 59510, Mexico; zochoac1900@alumno.ipn.mx (Z.O.-C.); vangoa@ipn.mx (M.V.A.-P.); jcardenasv@ipn.mx (J.G.C.-V.); garinacho@gmail.com (I.G.-R.); jceja@ipn.mx (J.A.C.-D.); jose.osvaldo.ber95@gmail.com (J.O.B.-G.); 2Laboratory of Phytobiochemistry, Departament of Biotechnology and Biochemistry, CINVESTAV IPN Unidad Irapuato, Irapuato 36821, Mexico

**Keywords:** hemiparasite, cyanidin-3-glucoside, *Prosopis laevigata*, *Quercus deserticola*, antioxidant, antimicrobial activity

## Abstract

*Psittacanthus calyculatus* is a hemiparasitic plant of an arboreal species (e.g., forest, fruit trees). Its foliage has therapeutic potential; however, little is known about its fruits. In this study, the phytochemical profile and biological activities of *P. calyculatus* fruits hosted by *Prosopis laevigata* and *Quercus deserticola* were evaluated. The fruits of *P. calyculatus* from *P. laevigata* showed the highest content of total phenols (71.396 ± 0.676 mg GAE/g DW). The highest content of flavonoids and anthocyanins was presented in those from *Q. deserticola* (14.232 ± 0.772 mg QE/g DW; 2.431 ± 0.020 mg C3GE/g DW). The anthocyanin cyanidin-3-glucoside was detected and quantified via high-performance thin-layer chromatography (HPTLC) (306.682 ± 11.804 mg C3GE/g DW). Acidified extracts from host *P. laevigata* showed the highest antioxidant activity via ABTS**^•+^** (2,2′azinobis-(3-ethylbenzothiazdin-6-sulfonic acid) (214.810 ± 0.0802 mg TE/g DW). Fruit extracts with absolute ethanol from the *P. laevigata* host showed the highest antihypertensive activity (92 ± 3.054% inhibition of an angiotensin converting enzyme (ACE)). Fruit extracts from both hosts showed a minimum inhibitory concentration (MIC) of 6.25 mg/mL and a minimum bactericidal concentration (MBC) of 12.5 mg/mL against *Escherichia coli*, *Salmonella choleraesuis* and *Shigella flexneri*. Interestingly, a significant host effect was found. *P. calyculatus* fruits extract could be used therapeutically. However, further confirmation experiments should be carried out.

## 1. Introduction

*Psittacanthus calyculatus* is commonly known as Mexican mistletoe; it classified as a hemiparasite plant since it synthesizes its own chlorophyll, although it depends on its hosts to germinate and feed by taking vital substances from them through a modified root called the haustorium [1,2].

The seeds are dispersed by frugivorous birds, which deposit them on the branches of their hosts, to which they adhere by means of a mucilaginous substance called viscin [3]. *P. calyculatus* is the most widely distributed species within the phyla Loranthaceae, with a wide distribution from Baja California, Sonora and Tamaulipas from Mexico to northern Argentina; it also has a wide range of hosts, such as trees, shrubs and cacti, both gymnosperms and angiosperms, among which are *Quercus* spp. (oak), *Acacia* spp., *Juglans* spp. (walnut), *Ficus* spp., *Populus* spp., *Salix* spp. (willow), *Prosopis* spp. (mesquite), *Prunus* spp. (peach, capulin) and *Persea americana* [1,3,4]. Due to its great affinity with forest and fruit trees, Mexican mistletoe is considered a weed, so there is interest in developing sustainable management strategies, since existing ones (e.g., pruning, herbicides) are neither sufficient nor effective [4,5,6,7].

*P. calyculatus* has been used as a medicinal plant in traditional Mexican medicine, applied topically or by means of infusions to treat ailments such as fevers, infertility or diabetes, among other conditions [8,9,10]. Recent studies in the area of pharmacology have reported that its foliage has coadjuvant effects in the treatment of arterial hypertension, since the substances present in its foliage act as vasodilators, lower cholesterol and blood sugar levels; they also display a cytotoxic effect against cancer cells [11,12,13,14].

Studies on other genera or species of the Loranthaceae family have shown the therapeutic potential of the different tissues of parasitic species [15,16,17]. Serrano et al. [15] reported the content of phenolic compounds and antioxidant capacity of the colored fruit of *Cladocolea loniceroides*, a species endemic to Mexico.

Regarding the other species of the genus *Psittacanthus*, which are endemic to other countries, the antimicrobial activity of *P. linearis* fruits, as well as the phytochemical content and antioxidant capacity of the leaves of *P. plagiofilus* and *P. cucullaris* have been reported, using different extractive solvents such as ethanol and water [18,19,20]

Therefore, the objective of this study was to determine the phenolic content and biological activities of the ripe fruits of *P. calyculatus*, since there are no studies demonstrating its potential benefits whose knowledge could contribute to the proposal of sustainable strategies for its control, as well as to the diversification of sources of secondary metabolites as the active ingredients of drugs and food supplements, among other things.

## 2. Results and Discussion

### 2.1. UV–Visible Spectrophotometry Analysis

#### 2.1.1. Total Phenol Quantification

A significant effect of the host *P. laevigata* and the extractive solvent on the TPC was observed (Table 1). Thus, the highest content of total phenols in fruits was obtained when the host was *P. laevigata* and the solvent used was acidified ethanol, being between 80.3% and 108% higher than the rest of the extracts evaluated.

There is a study on other parasitic species, *P. plagiophyllus*, in which an ethanol–water combination was used in different proportions as extractive solvent to achieve a higher yield [18]. This is the first phytochemical study of *P. calyculatus* that performs a phytochemical analysis of the fruit. Serrano et al. [15], evaluated the phenol and flavonoid content of *C. loniceroides* (Loranthaceae family), and the authors reported higher values of fruit total polyphenol content (189.5 mg EAG/g DW) than those obtained in this study (71.396 mg EAG/g DW).

#### 2.1.2. Total Flavonoids Quantification

In contrast to the total phenol content, the TFC of the host–host interaction and extractive solvent showed a significant effect on the flavonoid content of these extracts. On the other hand, the highest TFC was shown via the extracts with acidified ethanol when the host was *Q. deserticola*, with a higher content (14.232 EQ mg/g DW) being between 25% and 29.5% higher than the rest of the extracts evaluated (Table 1).

Studies of parasitic species of the Loranthaceae family have been carried out on the foliage, identifying flavonoids such as rutin, quercetin and hesperidin, as well as the presence of alkaloids, anthraquinones and steroids [12,13,18,19].

The study performed by Serrano et al. [15], mentioned above, evaluated the flavonoid content of *C. loniceroides* fruits as well, and the authors found higher contents of these compounds in the pigmented fruit (36.45 mg Rutin E/g DW) than in other parts of the plant. The reported values were higher than those found in the present work, which could be due to the use of rutin as a standard instead of quercetin.

#### 2.1.3. Total Anthocyanins Content

The highest TAC was observed in fruits from *P. calyculatus* when the host was *Q. deserticola* and the solvent used was ethanol, being between 219% and 196% higher than the rest of the extracts. In general, a higher recovery of metabolites was achieved using acidified ethanol (Table 1).

This is the first study to date to report on the total anthocyanin content of *P. calyculatus* fruits; existing studies address the phytochemistry of leaves or unpigmented fruits belonging to other families or species [15,18,19,21]. However, the properties of pigmented berries such as strawberries, blueberries, blackberries, grapes and other red fruits have been extensively studied for their neuroprotective, anti-inflammatory and antihyperglycemic effects in various applications [22,23,24,25,26]. Therefore, it is relevant to investigate the phytochemistry and biological properties of *P. calyculatus* fruit, since it could be a potential source of metabolites of interest, which in turn would promote the use and conservation of important forest resources [2,5,27].

It is important to note that certain physical and chemical modifications that the parasite undergoes have been reported by several authors, suggesting that these are the result of the influence of various biotic or abiotic factors, and especially emphasizing that the amount and content of metabolites depends mostly on the host [2,5,28,29].

It seems that the parasitic organism is dependent on its host to acquire the nutrients necessary for its survival, triggering a chemical host response, which, in this case, added value to the *P. calyculatus* fruit when considering the host *P. laevigata* as a promoter of the accumulation of fruit phenolic compounds and *Q. deserticola* as a promoter of the accumulation of fruit flavonoids and anthocyanins.

#### 2.1.4. Antioxidant Activity

A significant effect of the host, the extractive solvent and their interactions was found on the antioxidant capacity of ethanolic extracts of *P. calyculatus* fruits determined via the DPPH^●^ method, while the antioxidative capacity determined using the ABTS**^•+^** method was significantly influenced through the host and extractive solvent (Table 2).

In the DPPH^●^ assay, the highest activity was shown via the acidified extracts of both hosts. The ABTS**^•^**^+^ assay showed a significantly higher antioxidant activity in the acidified extract of the host *P. laevigata* (141.6–104.7%) compared to the rest of the extracts.

The host species of *P. calyculatus* showed no significant effect on the antioxidant activity of the ethanolic extracts of its fruits, determined via the ABTS**^•+^** method, while the extractive solvent significantly influenced this biological activity in the extracts measured via the DPPH^●^ method.

No studies were found documenting the antioxidant activity of *P. calyculatus* fruits. However, the antioxidant activity of leaves collected from different hosts (*Hedera helix*, *Fraxinus uhdei* and *P. laevigata*) was reported by Zarza et al. [11]. The authors observed a higher antioxidant activity in the aqueous extracts of leaves from specimens hosted by *P. laevigata* (91.01% inhibition in the DPPH^●^ assay). In addition to detecting a greater diversity of phenolic compounds (e.g., (+)-catechin, gallic acid and p-coumaric acid).

The differences found in the activity of the different extracts could be due to the chemical changes promoted by biotic and abiotic stimuli, as discussed above.

#### 2.1.5. MIC and MBC of Ethanolic Extracts of Pericarp of *P. calyculatus*

The lowest MIC and MBC values of the *P. calyculatus* fruit pericarp (Table 3) was shown via the acidified ethanol extracts for both hosts, *P. laevigata* and *Q. deserticola*. MIC and MBC were the same for all three enteropathogenic bacteria (*Salmonella, Escherichia* and *Shigella*). Antimicrobial studies on the genus *Psittacanthus* exist; however, they have not evaluated different hosts. Bailladores et al. [20] reported the antimicrobial activity of the pigmented fruit of the genus *P. linearis* in relation to *S. aureus*, *E. coli* and *Pseudomonas aeruginosa*, showing that only the ethanolic extract of the fruit was more active against the bacteria *S. aureus*, being more sensitive to its metabolites content (polyphenols with a concentration of 300 µg/mL), while the ethanolic extracts of the *P. calyculatus* fruit evaluated in the present study showed the same inhibitory and bactericidal activity against *S. choleraesuis* ATCC 10708, *E. coli* ATCC 12792, and *S. flexneri* ATCC 12022, observing that these bacteria were more sensitive to the total content of anthocyanins, followed by the content of total phenols, based on the correlation analysis (shown in Section 2.3).

There are no reports on the antimicrobial activity of *P. calyculatus* fruits; however, the leaf extracts of the *P. linearis* and *P. cucullaris* species have been evaluated against Gram-negative and Gram-positive bacteria (*P. aeruginosa*, *S. aureus* and *E. coli*), resulting in an inhibitory effect on these microorganisms [19,20].

#### 2.1.6. Antihypertensive Activity

The highest inhibition of ACE was observed in the absolute ethanol extract of fruits from both hosts (Table 4), significantly exceeding the acidified ethanol extracts of *P. laevigata* and *Q. deserticola* fruits (376% and 43.8%, respectively).

There is a patent reported by Cervantes et al. [27] about the antihypertensive properties of aqueous extracts of *P. calyculatus* leaves, which described the vasorelaxation tests performed on male guinea pigs of the Hartley strain, showing relaxation levels of 66–92% in the aortic rings. The results of this work also showed the antihypertensive potential of the *P. calyculatus* fruit from the same host (*P. laevigata*) through ACE inhibition (92.59%).

There is another study on the vasomotor responses of the aortic rings of roots, using the ethanol–water extraction of leaves of the mistletoe *P. calyculatus*, collected on *Quercus candicans*, in which the authors concluded that *P. calyculatus* had an effect on the endothelium, inducing the synthesis and release of nitric oxide at concentrations of 800 μg/mL [30].

There are several studies on the medicinal properties of other African parasitic plants, such as *Loranthus micranthus* and *Loranthus bengwensis*, which suggest that their ACE inhibition effect is host-dependent [31,32,33,34,35].

### 2.2. HPTLC

#### 2.2.1. Detection of C3G in Ethanolic Extracts of Pericarp of *P. calyculatus*

Regarding the detection and quantification of bioactive metabolites, C3G, whose chemical structure provides a dark coloration of purple-black tones to grapes, blackberries and blueberries, among other pigmented fruits, was detected via HPTLC and TLC [36,37,38]. Figure 1 shows the chromatogram of *P. calyculatus* fruit pericarp extracts: absolute ethanol, acidified ethanol and the C3G standard. A retention factor (Rf) value of 0.38 was obtained, and a coloration similar to that of the anthocyanin standard was used.

Regarding the quantification of C3G (Table 5), a higher recovery of the metabolite was obtained with the acidified ethanol extracts of pericarp of *P. calyculatus* fruits from both hosts (*P. laevigata* and *Q. deserticola*). The acidified extracts of fruits from specimens collected from the host *Q. deserticola* were almost double the concentration of C3G (306.682 mg C3GE/g DW) of the rest of the extracts.

There are studies of plants leaves of other genera and species belonging to the Loranthaceae family, such as *Plicosepalus curviflorus*, *Plicosepalus acacia*, *Helicanthus elastica* and *Scurrula atropurpurea*, among others, in which quercetin, catechin, flavonol gallate, alkaloids, anthra-quinones, terpenoids, tannins and quinones have been identified via HPTLC and TLC [39,40,41,42,43]. However, this is the first chromatographic study of the fruit of *P. calyculatus*, since the chromatographic reports found on this species concern the leaves, generally using the HPLC technique [13].

#### 2.2.2. HPTLC–DPPH^•^ Antioxidant Assay

As for the detection via HPTLC–DPPH of compounds with antioxidative potential (Figure 2), bands coinciding with the retention factor of the C3G (Rf = 0.380) were found.

Similar results were obtained in a study on the phytotherapeutic potential of berries determined via their antioxidant capacity (blue-berried honeysuckles), in which C3G showed a strong signal upon derivatization with DPPH**^•^** [36]. It is worth mentioning that C3G has been identified in pigmented fruits [44,45].

### 2.3. Pearson’s Correlation of the Analyses Evaluated

Pearson’s correlation (Table 6), together with the two-way statistical analyses, suggested that both the extractant solvent and the host have a significant effect on the recovery of metabolites from *P. calyculatus* fruit correlating with biological activities, such as antioxidant, antimicrobial and antihypertensive activities. MIC and MBC showed a desirable high negative correlation with TAC, TPC and C3G. In the same way, ACE obtained a high negative correlation with TPC and TAC, in addition to showing a high positive correlation with MIC and MBC. Antioxidant activity identified using the ABTS^●+^ method was highly correlated with TPC, while via the DPPH^●^ method it was highly correlated with TFC and C3G content.

## 3. Materials and Methods 

HPLC-grade methanol, hydrochloric acid (HCl), chlorogenic acid, gallic acid, quercetin, Trolox, anthocyanin standards in the form of chloride salts, thiazolyl blue tetrazolium blue (MTT), benzenesulfonyl chloride (BSC), hippuryl-L-histidyl-L-leucine (HHL) and angiotensin converting enzymes (ACEs) were purchased from Sigma-Aldrich^®^ (St. Louis, MO, USA). Silica gel plates 60 F254 for HPTLC (20 × 10 cm, Art. 1.05729.0001) were supplied by Merck^®^ (Darmstadst, Germany). For mobile phases, toluene, ethyl acetate, acetic acid and formic acid supplied by Sigma-Aldrich^®^ (St. Louis, MO, USA) were used. Broth and Mueller–Hinton agar were purchased from BD Bioxon (State of Mexico, Mexico). Certified strains of *Escherichia coli* ATCC 12792, *Salmonella choleraesuis* ATCC 10708 and *Shigella flexneri* ATCC 12022 were used.

### 3.1. Plant Samples

The mature fruits of *P. calyculatus* were collected on *Q. deserticola* and *P. laevigata* hosts in the localities of the La Arena municipality of Marcos Castellanos, Michoacán (19°59′25″ N, 102°53′57″ W) and Jiquilpan de Juárez, Michoacán (20°1′36″ N, 102°41′24″ W), in the autumn (October 2020). The plants were identified by M.S. Ignacio García Ruíz (collection number 9971 and 9974). Specimens were deposited in the CIIDIR—Michoacán Herbarium (CIMI).

### 3.2. Extract Preparation

The fruits were rinsed with distilled water, the pericarp was separated, lyophilized (Lyophilizer, LABCON, Kansas City, MO, USA) and stored at −20 °C until use. The pericarp of fruits from each host, i.e., oak and mesquite, was pulverized with a pestle and mortar. Extracts were prepared with absolute ethanol and ethanol acidified with 0.01% HCl. Samples of 1 g were extracted in 75 mL of extractive solvent via three washes. The extracts were sonicated at 55 ± 5 Hz (UL-TRAsonik, DENSTPLY, NEYTECH, Yucaipa, CA, USA) for 30 min at room temperature and left in agitation. The extracts were vacuum filtered through a 0.45 µm cellulose filter and stored at −20 °C.

### 3.3. Total Phenols Quantification

The quantification of total phenols (TPC) in the extracts was based on a calibration curve, as reported by Spinardi et al. [46]. The calibration curve A_760_ = 0.6032 [gallic acid] + 0.0404, *R*^2^ = 0.9929, was obtained from 10 concentrations (0.25–2.5 mg/mL) of gallic acid. The absorbance was measured at 760 nm using a spectrophotometer (PowerWave HT, Biotek, Biotek Instruments, Winooski, VT, USA). The total content was expressed in milligram gallic acid equivalents per gram of dry weight (mg GAE/g DW). The assay was performed in triplicate.

### 3.4. Total Flavonoids Quantification

Total flavonoid content (TFC) was determined using the technique reported by Woisky and Salatino [47] with some modifications, using a standard curve for quercetin, (A_425_ = 1.1113 [quercetin] − 0.0009, *R*^2^ = 0.9965) obtained from 10 concentrations (0.1–1 mg/mL). Absorbance was measured at 425 nm in the spectrophotometer (PowerWave HT, Biotek, Biotek Instruments, Winooski, VT, USA). The total content was expressed as milligram quercetin equivalents per gram of dry weight (mg QE/g DW). The assay was performed in triplicate.

### 3.5. Total Anthocyanin Quantification

The total anthocyanin content (TAC) was determined using the technique described by Abdel and Hucl [48]. The absorbance was measured at 535 nm with the spectrophotometer (PowerWave HT, Biotek, Biotek Instruments, Winooski, VT, USA). Total anthocyanin content was expressed as a cyanidin-3-glucoside equivalent (mg C3GE/g DW) and calculated using the following formula:(1)$$Content\;of\;total\;anthocyanins\;=\;(\frac{A_{535\;nm}}{25,965})(449)(\frac{1}{extract\;concentration})(10^{6})$$

### 3.6. High Performance Thin Layer Chromatography (HPTLC)

#### 3.6.1. General Conditioning

The identification and quantification of secondary metabolites was performed using the techniques reported by Cretu et al. [49]. Silica gel plates 60 F 254 of 10 × 20 cm (Merck^®^, (Darmstadst, Germany) were used. Plates were activated by heating (TLC Plate Heater 3, CAMAG, Muttenz, Switzerland) at 100 °C for 3 min. After cooling to room temperature (22 °C ± 2), samples were applied (1 g/75 mL) with a sample applicator (ATS 4, CAMAG, Muttenz, Switzerland), and the plate was developed with 10 mL of solvent system in the automated development chamber (ADC 2, CAMAG, Muttenz, Switzerland) at 47% RH (relative humidity) (in equilibrium with a saturated solution of potassium thiocyanate KSCN) for 10 min. The plates were derivatized with a 1% natural products (NP) methanolic solution (2-aminoethyl diphenyl borate, Sigma-Aldrich^®^, St. Louis, MO, USA) reagent and derivatization was performed in the immersion device (Chromatogram Immersion Device, CAMAG, Muttenz, Switzerland) at an immersion speed of 5 cm/s with an immersion time of 1 s. After derivatization, the plate was dried for 3 min at 100 °C (TLC Plate Heater 3, CAMAG, Muttenz, Switzerland). Images of each plate were documented using a TLC Visualizer (CAMAG, Muttenz, Switzerland) under visible light, at UV 254 nm and UV 366 nm.

Results were processed via the VisionCATS version 1.4.7.2018 software (CAMAG, Muttenz, Switzerland).

#### 3.6.2. Identification and Quantification of Cyanidin-3-Glycoside

For the identification and determination of anthocyanins the techniques reported by Cretu et al. [38] were applied. The analysis was performed in triplicate; silica gel plates 60 F 254 of 10 × 20 cm (Merck^®^, Darmstadt, Germany) were used, the plates were conditioned (preheating at 100° C for 3 min) and then a sample applicator (ATS 4, CAMAG, Muttenz, Switzerland) was used. The C3G standard was loaded at different volumes 1. 5, 3, 6, 9 and 12 µL in order to obtain a calibration curve (y = −1.946 × 10^−15^x^2^ + 5.207 *×* 10^−8^x − 3.155 × 10^−2^, *R^2^* = 0.9999). The standard was prepared at a concentration of 1 mg/mL of methanol. The absolute and acidified ethanolic fruit pericarp extracts were prepared at a concentration of 1 g/75 mL, a volume of 2.5 µL was applied to samples extracted with absolute ethanol and a volume of 4 µL was applied to samples extracted with ethanol acidified. The bands were adjusted and applied, with a position of Y: 8 mm and X: 16.8 mm, with a length of 7 mm and a distance between lanes of 10.4 mm. A total of 2.0 µL of the extracts and standards was applied at a constant application rate (100 nL/s). Plate development was carried out in the automated development chamber (ADC 2, CAMAG, Muttenz, Switzerland) at 47% RH (in equilibrium with a saturated solution of potassium thiocyanate KSCN) for 10 min, the plate was developed with 10 mL of a solvent system comprising ethyl acetate:acetic acid:formic acid:water (10:1.1:1:1:2.3). The developed plate was visualized, and a digitized record was obtained under white light and visible light at 254 nm and 366 nm on the visualizer (TLC Visualizer, CAMAG, Muttenz, Switzerland). Subsequently, the plate was derivatized with the natural products (NP) reagent in the immersion device (Chromatogram Immersion Device, CAMAG, Muttenz, Switzerland) and dried in a plate heater (TLC Plate Heater 3, CAMAG, Muttenz, Switzerland) at 100 °C for 3 min.

#### 3.6.3. Antioxidant Activity by HPTLC-DPPH^●^

Antioxidant activity was determined via the HPTLC-DPPH^●^ (2,2-difenil-1-picrilhidrazilo) method according to Orsini et al. [50] with some modifications. Developed plates for the identification and quantification of anthocyanins as previously described were derivatized with DPPH^●^ reagent (0.2% in methanol) in the immersion device (Chromatogram Immersion Device, CAMAG, Muttenz, Switzerland) using a vertical velocity of 5 cm/s, for 3 s. The plates were dried at room temperature (22 ± 2 °C) in the dark for 30 min. The images obtained were documented using the TLC Visualizer (CAMAG, Muttenz, Switzerland) under visible light. The data obtained were processed with VisionCATS version 1.4.7.2018 software (CAMAG, Muttenz, Switzerland).

### 3.7. Biological Activities

#### 3.7.1. Antioxidant Activity by ABTS^•+^ Assay

The radical-scavenging capacity of the methanolic extracts was determined using the ABTS^•+^ method, as reported by Hosu et al. [51]. The ABTS solution was prepared by dissolving 360 mg of ABTS**^•+^** (2,2′azinobis-(3-ethylbenzothiazdin-6-sulfonic acid) in 100 mL of deionized water, and 100 mL of ABTS solution was added to 100 mL of 2.45 mM potassium persulfate solution for the activation of ABTS^•+^. The mixture reaction was conserved in the dark for 24 h. The absorbance was measured at 734 nm using (PowerWave HT, Biotek, Biotek Instruments, Winooski, VT, USA) and was adjusted to about 0.760 ± 0.001 through dilution with distilled water. A total of 280 µL of ABTS solution was added to 20 µL of methanolic extracts in a 96-well microplate, and the mixture was kept in the dark for 15 min. The absorbance of the sample and the blank were measured at 734 nm using the spectrophotometer (PowerWave HT, Biotek, Biotek Instruments, Winooski, VT, USA). The antioxidant capacity of the extracts was expressed as μmoles Trolox equivalent/g DW (μmol TE/g DW) using a calibration curve (μmol TE = −5.4371 [A_734nm_] + 1.4357, *R*^2^ = 0.9852) at five different concentrations (0–0.60 μmol Trolox). The assay was performed in triplicate.

#### 3.7.2. Antioxidant Activity by DPPH^●^ Assay

The antioxidant activity of the extracts was determined according to the DPPH^●^ method described by Marín et al. [19] with some modifications. In 96-well microplates, 20 μL of the ethanolic and ethanolic acidified extracts diluted (1:4) and 200 μL of DPPH^●^ solution (150 mM, ethanolic solution) were added. The mixtures were incubated for 30 min in the dark. The absorbance was measured at 515 nm using the spectrophotometer (PowerWave HT, Biotek, Biotek Instruments, Winooski, VT, USA) and the antioxidative capacity of the extracts was expressed as the milligram equivalents of the antioxidant capacity of Trolox per gram dry weight (μM ET/g PS) using a calibration curve (μM ET = −2.8918 [A_515nm_] + 1.4357, *R*^2^ = 0.9849) at five concentrations (0–0.60 μmol Trolox).

#### 3.7.3. Determination of Antihypertensive Activity via ACE Inhibition

The assay was performed based on the hydrolysis of hippuryl-L-histidyl-L-leucine (HHL) vua ACE to hippuric acid (HA) and histidyl-leucine (HL) as products, as reported by Lia et al. [52], with some modifications. For each assay (sample, control and blank) according to Li et al. (2005), an 8 µL ACE inhibitor sample solution was prepared with 50 µL of 5 mM HHL in 100 mM sodium borate buffer (pH 8.3) containing 300 mM NaCl, which was previously incubated at 37 °C for 5 min. The reaction was initiated by the addition of 2 µL of ACE solution (100 mU/mL) and the mixture was incubated at 37 °C for 30 min. The reaction was halted by adding 40 µL of 1 M HCl, in conjunction with 30 µL of sodium borate. Subsequently, 96 µL of quinolone was added, followed by 32 µL of benzenesulfonyl chloride (BSC) and then incubation at 30 °C for 30 min in the dark. A buffer was then added to the mixture of 570 µL of absolute ethanol in each assay (sample, control and blank). Finally, followed by incubation at 30 °C for 30 min, absorbance was measured on a microplate reader (spectrophotometer, PowerWave HT, Biotek, Biotek Instruments, Winooski, VT, USA) at 492 nm. All measurements were performed in triplicate. A calibration curve versus HA concentration was obtained. Captopril (25 mg) was used as a control. The percentage calculation of ACE inhibitory activity was performed by applying the following equation:(2)$$ACE\;inhibitory\;activity\left(\%\right)=\frac{B-A}{B-C}\times 100$$
where A = sample; B = control; and C = blank.

#### 3.7.4. Determination of Antimicrobial Activity

The antimicrobial activity was determined according to that reported by Karuppasami et al. [53] with some modifications. The MIC and MBC were determined using the standard broth microdilution method. Extracts with concentrations of 50, 37.5, 25, 18.75, 12.5, 6.54 and 2.35 mg/mL were placed in sterile polystyrene 96-well microplates containing 100 μL of Mueller–Hinton broth. A 24 h inoculum of *E. coli* ATCC 12792, *S. choleraesuis* ATCC 10708 *S. flexneri* ATCC 12022) was adjusted to 1 × 10^7^ CFU/mL and 50 µL of each bacterial suspension were added. As controls, 200 μL of Mueller–Hinton broth and 20 μL of the inoculum adjusted to a density of 1 × 10^7^ CFU/mL were used. The plates were incubated at 37 °C for 19 h and 20 μL of MTT tetrazolium salts were added before continuing the incubation for 45 min; finally, the MIC was recorded as the lowest concentration of the extracts without visible bacterial growth. For the determination of the MBC, as the lowest concentration that killed 100% of the bacteria, an aliquot of 50 μL from the wells without coloring was spread on plates with Mueller–Hinton agar medium and incubated at 37 °C for 24 h. The following controls were considered: positive control, broth + bacteria; negative control, broth only; and antibiotic control, broth + bacteria + antibiotic (30 µg ciprofloxacin). Three replicates of each of the dilutions and controls were performed. The tests were performed in triplicate on different dates.

### 3.8. Statistical Analysis

The results were reported with the mean ± standard deviation. A two-way analysis of variance (ANOVA) was performed, the separation of means was performed using Tukey’s test (*p* < 0.05) and Pearson’s correlation was also carried out. R Studio software version 4.0.3. was used.

## 4. Conclusions

The phytochemical and bioactive properties of the ripe fruits of *Psittacanthus calyculatus* were studied here for the first time. The cyanidin-3-glucoside was detected in ethanolic extracts (with absolute and acidified ethanol), which showed antihypertensive potential, antioxidant capacity and antibacterial activity (against *S. choleraesuis*, *E. coli* and *S. flexneri*). Moreover, it was observed that, by means of acidified ethanolic extraction, a greater recovery of metabolites was obtained from the fruit.

The results showed that the host was determinant for the accumulation of secondary metabolites. An effect of the hosts on the phenolic content of *P. calyculatus* fruits was found, the host *P. laevigata* favored the accumulation of phenolic compounds, while *Q. deserticola* promoted a higher content of flavonoids and anthocyanins. However, further research on the interaction of *P. calyculatus* and its hosts is needed to clarify this phenomenon.

Based on the results obtained, a range of potential benefits for ethnopharmacology, industry and agriculture, as well as other sectors, can be envisaged.

## Figures and Tables

**Figure 1 plants-12-02292-f001:**
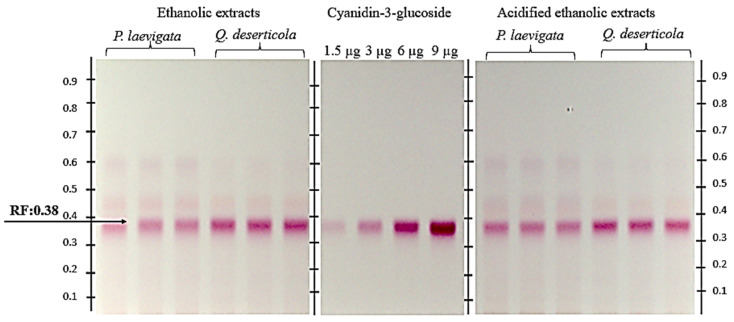
Chromatogram of pericarp samples of mature *P. calyculatus* fruit extracted with absolute ethanol from two different hosts, namely *P. laevigata* (Mesquite) and *Q. deserticola* (Oak). Standard C3G was applied at different volumes with four calibration points, i.e., 1.5, 3, 6 and 9 µL, equivalent to 1.5, 3, 6 and 9 µg; y = −1.946 × 10^−15^x^2^ + 5.207 × 10^−8^x − 3.155 × 10^−2^, *R^2^* = 0.9999. In a sample application volume of 2 µL (26.66 DW /mL), in triplicate. Pericarp samples of *P. calyculatus* extracted with acidified ethanol with respect to two distinct hosts, *P. laevigata* (Mesquite) and *Q. deserticola* (Oak) (viewed under white light, derivatized with NP (Sigma-Aldrich^®^, St. Louis, MO, USA) under a solvent system of ethyl acetate:acetic acid:formic acid:water (10:1.1:1.1:2.3). Retention factor (Rf) = 0.38.

**Figure 2 plants-12-02292-f002:**
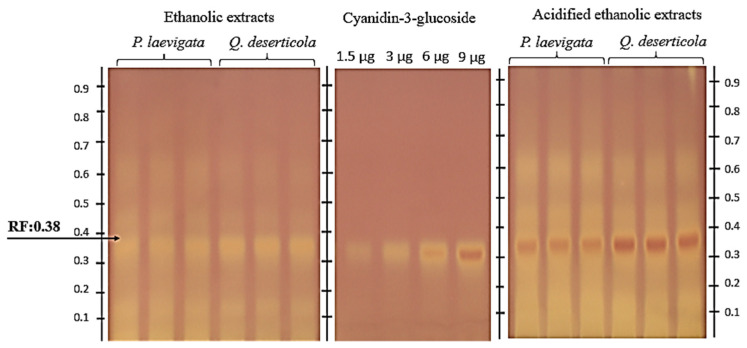
Chromatogram of pericarp samples of *P. calyculatus* extracted with absolute ethanol with respect to two different hosts *P. laevigata (*Mesquite) and *Q. deserticola (*Oak). Standard C3G was applied at different volumes with four calibration points, namely 1.5, 3, 6 and 9 µL, equivalent to 1.5, 3, 6 and 9 µg; y = −1.946 × 10^−15^x^2^ + 5.207 × 10^−8^x − 3.155 × 10^−2^, *R^2^* = 0.9999. In a sample application, a volume of 2 µL (26.66 DW /mL), in triplicate, acidified the ethanol extracts of two distinct hosts, *P. laevigata* (Mesquite) and *Q. deserticola* (Oak)*,* viewed under white light and a DPPH**•**-derived anthocyanin assay plate.

**Table 1 plants-12-02292-t001:** Total phenolic, flavonoid and anthocyanin content of ethanolic extracts of *P. calyculatus* fruit pericarp from two different hosts.

Host (H)	Extractive Solvent (ES)	Total Phenols (mg GAE/g DW)	Total Flavonoids (mg QE/g DW)	Total Anthocyanins (mg C3GE/g DW)
*P. laevigata*	Acidified ethanol Ethanol	71.396 ± 0.676 ^a^ 39.588 ± 0.394 ^d^	10.986 ± 0.486 ^b^ 12.438 ± 0.758 ^b^	2.209 ± 0.032 ^b^ 0.826 ± 0.003 ^c^
*Q. deserticola*	Acidified ethanol Ethanol	64.457 ± 0.522 ^b^ 34.179 ± 0.356 ^c^	14.232 ± 0.772 ^a^ 11.374 ± 0.505 ^b^	2.431 ± 0.020 ^a^ 0.765 ± 0.034 ^c^
	(ES)	*	*	*
	(H)	*	*	*
	Interaction (ES:H)	*	-	*

GAE = gallic acid equivalents; QE = quercetin equivalents; CGE = cyanidin-3-glucoside equivalents; DW = dry weight. Mean ± standard deviations are presented. Two-way analysis of variance (ANOVA) was performed. * Indicates significant difference. Tukey’s test (n = 3, *p* ˂ 0.05). SE:H interaction separation of means was performed (n = 3, *p* ˂ 0.05). Different superscript letters within the same column indicate significant differences.

**Table 2 plants-12-02292-t002:** Antioxidant capacity of ethanolic extracts of pericarp of *P. calyculatus* fruits from two different hosts.

Host (H)	Extractive Solvent (ES)	DPPH• Method (mg TE/g DW)	ABTS^•+^ Method (mg TE/g DW)
*P. laevigata*	Acidified ethanol Ethanol	47.505 ± 0.247 ^ab^ 46.595 ± 0.137 ^bc^	214.810 ± 0.802 ^a^ 151.602 ± 0.401 ^c^
*Q. deserticola*	Acidified ethanol Ethanol	48.197 ± 0.518 ^a^ 47.284 ± 0.299 ^c^	205.036 ± 0.201 ^b^ 120.922 ± 0.201 ^d^
	(SE)	*	*
	(H)	*	*
	Interaction (SE:H)	*	-

TE = Trolox equivalents. Mean ± standard deviations are presented. Two-way analysis of variance (ANOVA) was performed. * Indicates significant difference. Tukey’s test (n = 3, *p* ˂ 0.05). SE:H interaction separation of means was performed (n = 3, *p* ˂ 0.05). Different superscript letters within the same column indicate significant differences.

**Table 3 plants-12-02292-t003:** Antimicrobial activity of ethanolic extracts of *P. calyculatus* pericarp from two different hosts against *S. choleraesuis*, *E. coli* and *S. flexneri*.

Microorganisms	Host	Extract	MIC (mg/mL)	MBC (mg/mL)
*S. choleraesuis* *ATCC 10708*	*P. laevigata*	Acidified ethanol Ethanol	6.25 12.5	12.5 16.6
	*Q. deserticola*	Acidified ethanol Ethanol	6.25 12.5	12.5 16.6
*E. coli* *ATCC 12792*	*P. laevigata*	Acidified ethanol Ethanol	6.25 12.5	12.5 16.6
	*Q. deserticola*	Acidified ethanol Ethanol	6.25 12.5	12.5 16.6
*S. flexneri* *ATCC 12022*	*P. laevigata*	Acidified ethanol Ethanol	6.25 12.5	12.5 16.6
	*Q. deserticola*	Acidified ethanol Ethanol	6.25 12.5	12.5 16.6

Average values reported in mg dry weight/mL. MIC = minimum inhibitory concentration; MBC = minimum bactericidal concentration.

**Table 4 plants-12-02292-t004:** Antihypertensive activity of ethanolic extracts of the *P. calyculatus* fruit pericarp from two different hosts.

Host (H)	Extractive Solvent (ES)	% of ACE Inhibition
*P. laevigata*	Acidified ethanol	19.400 ± 3.0547 ^c^
	Ethanol	92.592 ± 2.645 ^a^
*Q. deserticola*	Acidified ethanol	64.373 ± 1.527 ^b^
	Ethanol	89.947 ± 2.645 ^a^
	(ES)	*
	(H)	*
	Interaction (ES:H)	*

ACE: inhibition of angiotensin converting enzyme. Mean ± standard deviations are presented. Two-way analysis of variance (ANOVA) was performed. * Indicates significant difference. Tukey’s test (n = 3, *p* ˂ 0.05). SE:H interaction separation of means was performed (n = 3, *p* ˂ 0.05). Different superscript letters within the same column indicate significant differences.

**Table 5 plants-12-02292-t005:** Quantification of anthocyanins from pericarp of *P. calyculatus* from two different hosts by HPTLC.

Host (H)	Extractive Solvent (ES)	C3GE mg/g DW
*P. laevigata*	Acidified ethanol	201.846 ± 8.784 ^b^
	Ethanol	118.384 ± 3.464 ^d^
*Q. deserticola*	Acidified ethanol	306.682 ± 11.804 ^a^
	Ethanol	163.455 ± 3.854 ^c^
	(ES)	*
	(H)	*
	Interaction (ES:H)	*

C3GE: cyanidin-3-glucoside equivalents. Mean ± standard deviation is presented. Two-way analysis of variance (ANOVA) was performed. * Indicates significant difference. Tukey’s test (n = 3, *p* ˂ 0.05). SE:H interaction separation of means was performed (n = 3, *p* ˂ 0.05). Different superscript letters within the same column indicate significant differences.

**Table 6 plants-12-02292-t006:** Pearson’s correlation of biological activities and quantifications of metabolites in the ripe fruits of *P. calyculatus*.

	TPC	TFC	TAC	C3G	DPPH^●^	ABTS^●+^	MIC	MBC	ACE
TPC	1	0.77	0.96	0.69	0.60	0.98	−0.98	−0.98	−0.90
TFC		1	0.83	0.89	0.88	0.68	−0.79	−0.79	−0.43
TAC			1	0.85	0.75	0.95	−0.99	−0.99	−0.78
C3G				1	0.86	0.66	−0.81	−0.81	−0.40
DPPH^●^					1	0.54	−0.72	−0.72	−0.43
ABTS^●+^						1	−0.96	−0.96	−0.84
MIC							1	1	0.84
MBC								1	0.84
ACE									1

TPC = total phenol content; TFC = total flavonoid content; TAC = total anthocyanin content; C3G = cyanidin-3-glucoside content; DPPH^●^ = antioxidant capacity DPPH^●^ method; ABTS^●+^ = antioxidant capacity ABTS^●+^ method; MIC = minimum inhibitory concentration; MBC = minimum bactericidal concentration; ACE = antihypertensive activity.

## Data Availability

Not applicable.

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
