# Peer review of "Phytochemical Analysis and Biological Activities of Ripe Fruits of Mistletoe (Psittacanthus calyculatus)"

_plants, 2023, doi:10.3390/plants12122292_

Round 1

Reviewer 1 Report

Overall the manuscript was interesting to read, however, there are a plethora of grammatical errors throughout. Hence, I would suggest that a professional editing service be sought prior to resubmission. Also, there are numerous references missing, specific instances are noted in my general comments below.

General comments:

Abstract:

-line 18: no need to capitalise the first letter of 'high performance thin layer chromatography'. Please amend.

-line 20: please define the term 'ABTS'.

-line 22: no need to capitalise the first letter of 'angiotensin converting enzyme. Please amend.

-lines 23 and 24: no need to capitalise the first letter of ' minimum inhibitory concentration' or 'minimum bactericidal concentration'. Please amend.

Introduction:

-line 35, sentence beginning with (SBW): 'The seeds', requires referencing, please amend.

-lines 36/7, SBW: 'P. calyculatus', requires referencing, please amend.

-line 48, sentence ending with: 'etc', please delete etc and never use the term in a scientific manuscript. Also, the sentence requires referencing, please amend.

-line 52, SBW: 'Studies on other genera', requires multiple referencing, please amend.

Materials and Methods:

-line 67: no need to capitalise 'thiazolyl blue tetrazolium blue'. Please amend.

-line 86: what percentage of ethanol was used? Please include details.

-line 102: by convention, any number <10 is written in full, whilst those greater than or equal to 10 are written numerically. Hence, please change 'ten' with '10'.

-line 116: were the plates cooled completely or to a specific temperature? Please clarify.

-lines 118 and 143: no need to capitalise 'automated development chamber.' Please amend.

-line 119: please define the term 'RH'.

-line 120: no need to capitalise 'natural products'. Please amend.

-lines 152/3: please define the term 'DPPH'.

-line 191: no need to capitalise 'sample'. Please amend.

-line 198: what was the buffer comprised of and what was its pH? Please include details.

-lines 208 and 209: the terms MIC and MBC have already been defined (lines 23 and 24). Please only use abbreviations here.

-line 216: 'MTT' has already been defined. Please only use abbreviation here. Also, please check your materials and methods lines 66-75 for various abbreviations and ensure that from line 76 onwards, only the abbreviations are used and not redefined again.

Results and Discussion:

-line 237, SBW: 'There are studies' requires multiple referencing, please amend.

-line 239, SBW: 'On the other hand' the sentence is far too long. Please amend and rewrite into a minimum of two sentences.

-lines 256-262, 288-294, 313-319, 348-357: sentences are far too long. Please amend.

-line 322: What does the term MIB refer to? Shouldn't it be MBC? Please amend accordingly.

-line 330: please abbreviate Staphylococcus aureus to S. aureus as you have already define the bacterium on line 328.

-line 333: please abbreviate Escherichia coli to E. coli as you have already define the bacterium on line 328/9.

-line 369: please italicise the plant name.

-line 387, SBW: 'In the acidified' is grammatically incorrect, please amend.

-line 417: please delete 'On the other hand'. (Also, this phrase is grossly overused in this manuscript. Please amend accordingly.

Conclusion:

The whole conclusion requires editing (especially the first paragraph). Please amend.

As I stated above, I would advise the authors to seek the services of a professional editor prior to resubmission.

Reviewer 2 Report

The study evaluated the phytochemical profile and biological activities of the fruits of Psittacanthus calyculatus, a hemiparasitic plant found in arboreal species. The fruits, hosted by Prosopis laevigata and Quercus deserticola, were analyzed. 

This work could be published but needs significant revisions listed below.

The authors use the HPTLC technique instead of HPLC analysis. HPLC  has several advantages over HPTLC, including higher sensitivity, better resolution, greater accuracy and precision, greater reproducibility, and the ability to analyze a wider range of compounds. Please can the authors explain why the HPTLC technique is preferable to HPLC?

In paragraph 2.5 (line 108) the authors reported that they used a technique described in the reference [17]. Unfortunately, in this reference, there is no method for anthocyanine determination. Please could the authors clarify this issue?

Paragraph 2.6.1 e 2.6.2 

The authors reported that they used the techniques cited in the reference [18] (lines 114 and 132) regarding the identification and quantification of secondary metabolites (line 113) and Cyanidin-3-Glycoside (line 131). Unfortunately, in the reference [18] no HPTLC techniques were reported, and Cyanidin-3-Glycoside was not mentioned. Please, could the author clarify exhaustively about these incongruences?

In addition, in paragraph 2.6.2, it is reported a quadratic calibration curve without R-squared (R2).  Again in paragraph 2.6.2 it is reported a mobile phase (line 146) where the formic acid is reported twice! Please, can you correct the mobile phase used? In the same line please could the authors correct the volumes proportion of the components of the solvent system adopted.

Finally, I recommend that the author review the entire manuscript to verify the correspondence of the bibliographic references reported with the text.

Round 2

Reviewer 2 Report

The authors have addressed all the review comments. However remains the fact that the authors utilized an  HPTLC technique instead of HPLC analysis. If this in accordance with the journal editorial lines, the paper can be pubiished.
